# Amorphous (lysine)$_2$PbI$_2$ layer enhanced perovskite photovoltaics

Yehui Wen[1,2,8], Tianchi Zhang[1,2,8], Xingtao Wang[3,8], Tiantian Liu[4,8], Yu Wang[5], Rui Zhang[5], Miao Kan[6], Li Wan[7], Weihua Ning[2]✉, Yong Wang[1]✉ & Deren Yang[1]✉

Passivation materials play a crucial role in a wide range of high-efficiency, high-stability photovoltaic applications based on crystalline silicon and state-of-the-art perovskite materials. Currently, for perovskite photovoltaic, the mainstream passivation strategies routinely rely on crystalline materials. Herein, we have invented a new amorphous (lysine)$_2$PbI$_2$ layer-enhanced halide perovskite. By utilizing a solid phase reaction between PbI$_2$ and lysine molecule, an amorphous (lysine)$_2$PbI$_2$ layer is formed at surface/grain boundaries in the perovskite films. The amorphous (lysine)$_2$PbI$_2$ with fewer dangling bonds can effectively neutralize surface/interface defects, achieving an impressive efficiency of 26.27% (certified 25.94%). Moreover, this amorphous layer not only reduces crystal lattice stress but also functions as a barrier against the decomposition of organic components, leading to suppressed de-structuring of perovskite and highly stable perovskite solar cells.

State-of-the-art perovskite solar cells (PSCs) are emerging next-generation photovoltaic (PV) technology, achieving a certified efficiency of ~26.1% and experiencing active commercialization[1–3]. The progression in perovskite PVs strongly relies on the development of a new passivator, pivotal in enhancing the performance of PSCs[4]. The state-of-the-art passivation materials are essential for achieving superior efficiency and stability in PV applications, whether for silicon-based or perovskite-based PVs[5,6]. In particular, the perovskite absorbing layer in the PSCs is easily destructed under operational conditions due to the weakly bonded organic cations (like MA$^+$: Methylammonium and FA$^+$: Formamidinium) and inorganic [PbI$_6$]$^{4-}$ octahedral framework[7,8]. Therefore, a series of crystal passivators, such as organic molecules (PEAI, lysine, etc.) and low-dimensional perovskite (PEA$_2$PbI$_4$, etc.), have been developed to improve the PSCs' performance[9–12]. Among these, amino acid-based materials stand out as excellent passivate materials[13,14]. The organic groups such as

-COOH/-NH$_2$ in these molecules can passivate the unsaturated ion/vacancy defects on the surface/interface of perovskites. For example, lysine was used to surface ligands to chemically interact with unsaturated Pb on the surface of perovskite[15,16]. In all, these passivation strategies for efficient perovskite PV have consistently relied on the use of these two categories of materials (Table 1). These crystalline passivations with anisotropic behavior, due to the inherent characteristics of their crystal structures, inevitably exhibit limitations[17,18]. For example, crystal passivation easily results in compatibility issues when they interface with polycrystalline perovskite films. The robust ion-bonding effect between the organic molecule and perovskite, or the mismatch lattice between low-dimensional materials and perovskite, can lead to increased lattice stress, ultimately deteriorating long-term performance[19–21]. In addition, the interface between polycrystalline perovskite and crystal passivating materials typically hosts numerous dislocations, dangling bonds, and lattice strain, all of which

[1]State Key Laboratory of Silicon and Advanced Semiconductor Materials and School of Materials Science and Engineering, Hangzhou Global Scientific and Technological Innovation Center, Zhejiang University, Hangzhou, Zhejiang, P. R. China. [2]Institute of Functional Nano & Soft Materials, Joint International Research Laboratory of Carbon-Based Functional Materials and Devices, Soochow University, Suzhou, P. R. China. [3]Huaneng Clean Energy Research Institute, Beijing, P. R. China. [4]School of Chemistry and Chemical Engineering, Xi'an University of Architecture and Technology, Xi'an, P. R. China. [5]Department of Physics, Chemistry, and Biology (IFM), Linköping University, Linköping, Sweden. [6]School of Chemistry and Molecular Engineering, East China University of Science and Technology, Shanghai, P. R. China. [7]Max Planck Institute of Microstructure Physics, Halle, Germany. [8]These authors contributed equally: Yehui Wen, Tianchi Zhang, Xingtao Wang, Tiantian Liu. ✉e-mail: whning@suda.edu.cn; yonwa12@zju.edu.cn; mseyang@zju.edu.cn

degrade PV performance, particularly in terms of operation stability[22]. Inspired by commercial crystalline silicon PV, in which amorphous passivation technology is the key to achieving ultra-efficient PV performance[23,24], it becomes crucial to explore a strategy for perovskite PV that moves beyond anisotropic crystalline materials. Such innovation can further promote the potential performance of perovskite materials, especially in terms of stability.

Herein, we have developed an amorphous $(lysine)_2PbI_2$ passivator for highly efficient PSCs. The isotropic amorphous $(lysine)_2PbI_2$ with fewer dangling bonds not only effectively neutralizes surface/interface trap states and improves the excitonic quality of perovskite film, but also functions as a barrier against the decomposition of $MA^+/FA^+$ cations and enhance the perovskite's stability. Finally, the champion PSCs yield a high-power conversion efficiency (PCE) of 26.27% with improved open circuit voltage ($V_{oc}$) and fill factor (FF) compared with the reference device (23.72%). Moreover, the de-structuring of perovskite is suppressed under illumination and/or thermal coupling conditions, leading to excellent operational stability of the PSCs.

## Results and discussion
### The formation of amorphous $(lysine)_2PbI_2$

Optical and structural measurements of perovskite films reveal that the lysine reacts with residual $PbI_2$ within $FA_{0.85}MA_{0.1}Cs_{0.05}PbI_3$ (denoted as $PbI_2$-FACsMA) films and does not form any new crystal phases. The solid-state diffusion process of lysine into $PbI_2$-FACsMA perovskite film is shown in Supplementary Fig. S1 (the resulting films, denoted as amo-FACsMA), which is probed using ultraviolet-visible (UV-vis) spectroscopy. There is no notable change at the absorption edge of the amo-FACsMA perovskite film (Supplementary Fig. S2). Grazing-Incidence Wide-Angle X-ray Scattering (GIWAXS) is employed to assess the types of crystal phase in these perovskite films (Fig. 1a). This characterization reveals a distinct difference between $PbI_2$-FACsMA and amo-FACsMA films. The ring related to the $PbI_2$ phase is observed in the $PbI_2$-FACsMA film, while the ring associated with the $PbI_2$ phase is absent in the amo-FACsMA film. This strong contrast indicates $PbI_2$ disappearance without the emergence of a new crystal phase in amo-FACsMA film. To further understand this phenomenon, the crystal structure changes in amo-FACsMA film during the lysine diffusion process are monitored using X-ray diffraction (XRD) analysis (Supplementary Fig. S3). To rule out the impact of pressure and heat, the same process is conducted by employing a clean glass. The XRD pattern for $PbI_2$-FACsMA perovskite exhibits negligible changes, while a progressive reduction of $PbI_2$ is observed in the amo-FACsMA sample during the lysine diffusion process. Interestingly, a consistent pattern observed throughout the experiment is the disappearance of $PbI_2$, which

**Table 1 | Typical passivation materials currently used for high-efficiency perovskite PV**

| Component | Anisotropic passivating materials | | PCE/% | Reference |
|---|---|---|---|---|
| $FAPbI_3$ | Organic molecules | Hydrofluorocarbon | 26.07 | Nature, 2023, 623, 531 |
| $FA_{0.95}Cs_{0.05}PbI_3$ | | 1-(phenylsulfonyl)pyrrole | 26.1 | Nature, 2023, 624, 557 |
| $Cs_{0.05}MA_{0.15}FA_{0.8}PbI_3$ | | 3-fluoro-phenethylammonium iodide | 24.09 | Science, 2023, 38, 209 |
| $FA_{0.88}MA_{0.07}Cs_{0.05}PbI_{2.89}Br_{0.11}$ | Low-dimensional Materials | $(Cl_4Tm)_2PbI_4$ | 24.6 | Sci. Adv., 2023, 9, eadg0032 |
| $Cs_{0.05}FA_{0.95}PbI_3$ | | $(Al)_2(FA)_{n-1}Pb_nI_{3n-1}Cl_2$ | 25.12 | Nat. Energy, 2023, 8, 946 |
| $Cs_{0.15}FA_{0.85}PbI_{2.8}Cl_{0.2}$ | | $PEA_2ZnX_4$ | 24.1 | Nat. Energy, 2023. 8, 284 |

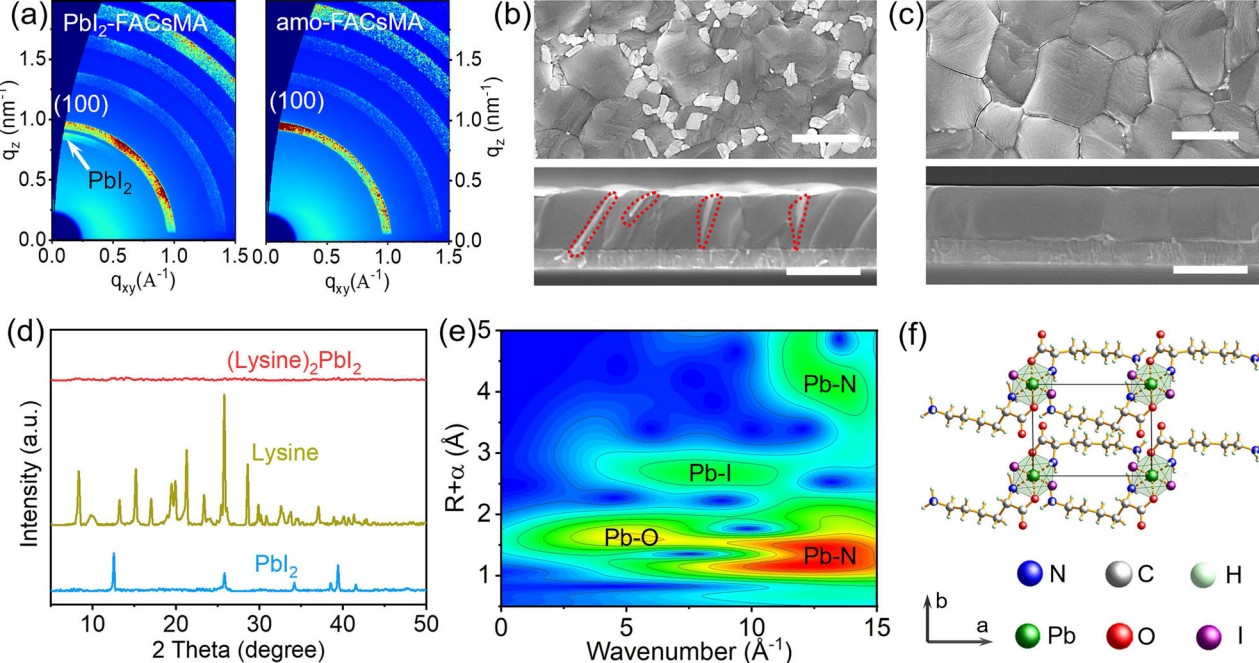

**Fig. 1 | Structural, morphological, and spectroscopic characterizations of amo-FACsMA films. a** GIWAXS data of $PbI_2$-FACsMA and amo-FACsMA films. Top-surface and cross-section images of (**b**) $PbI_2$-FACsMA and (**c**) amo-FACsMA perovskite films, the dashed box represents $PbI_2$. Scale bars are 1 μm. **d** XRD patterns for $PbI_2$, Lysine, $(Lysine)_2PbI_2$, a. u., arbitrary units. **e** Wavelet transformations-EXAFS images of $(Lysine)_2PbI_2$. **f** Schematic of amorphous $(lysine)_2PbI_2$.

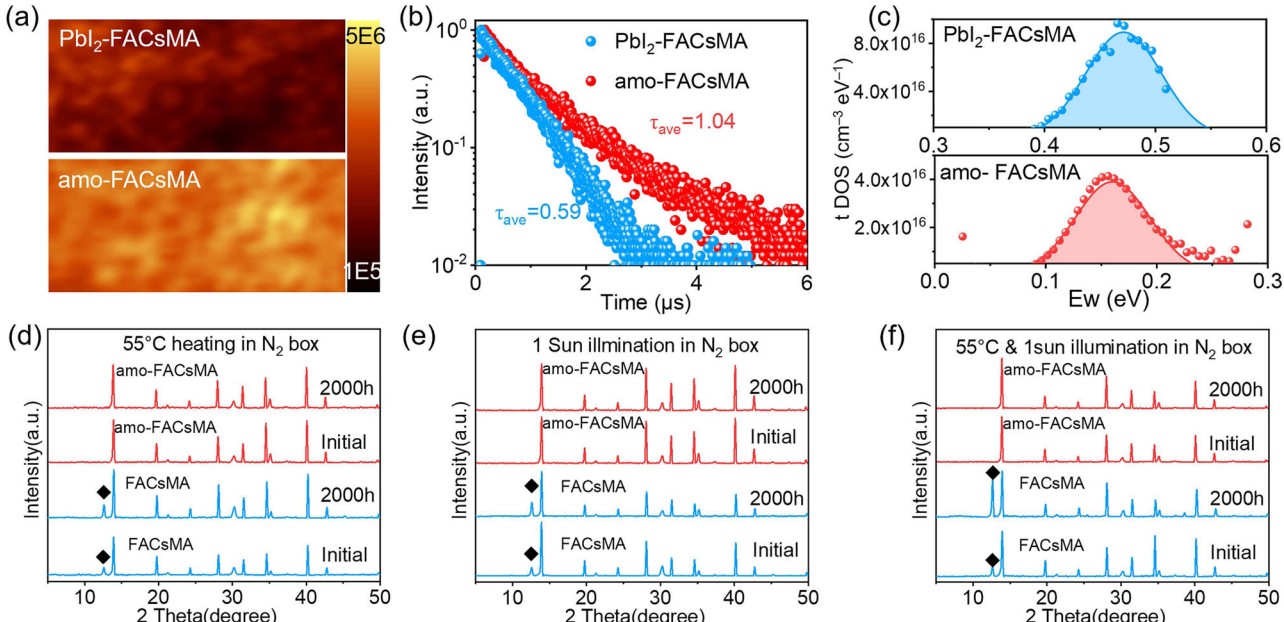

**Fig. 2 | Effect of isotropic (lysine)$_2$PbI$_2$ on the performance of perovskites. a** PL mapping results for PbI$_2$-FACsMA and amo-FACsMA films (area: 10 μm × 5 μm), respectively. **b** TRPL curves of PbI$_2$-FACsMA and amo-FACsMA films. **c** Density of states of traps (tDOS) deduced from the room-temperature C−f plots for PbI$_2$- FACsMA and amo-FACsMA. XRD results of perovskite films after (**d**) 55 °C heating, (**e**) 1 sun illumination, and (**f**) 55 °C heating + 1 sun illumination stability test in a glove box, the diamond represents PbI$_2$.

occurs without the formation of any new crystal structures, aligning with the results from GIWAXS.

Morphological and composition studies further indicate that the lysine reacts with the PbI$_2$ phase and potentially forms an amorphous material, which improves the film morphologies. PbI$_2$-FACsMA film shows substantial PbI$_2$ residues, while amo-FACsMA film shows pinhole-free grains without PbI$_2$ at both surface and grain boundaries (GBs) (Fig. 1b, c). High-resolution SEM images reveal evidence of solid-phase reactions for the PbI$_2$ residues in PbI$_2$-FACsMA film during the lysine diffusion process (Supplementary Figs. S4, S5). Time-of-Flight Secondary Ion Mass Spectrometry (TOF-SIMS) is conducted to verify the distribution of lysine within these perovskite films (Supplementary Fig. S6). In the amo-FACsMA perovskite films, lysine exhibits a vertical distribution similar to that of PbI$_2$ in the cross-section profile of Fig. 1b, predominantly accumulated at the upper surface layer. The addition of lysine causes minimal changes in the distribution of FA$^+$, MA$^+$, Cs$^+$, Pb$^{2+}$, and I$^-$ (Supplementary Fig. S7). Overall, the alterations in morphology and composition, along with the absence of new crystal phases in GIWAXs and XRD, suggest the formation of an amorphous material in the surface/interface of amo-FACsMA perovskite[25].

Spectroscopy and structure investigations confirm that lysine can react with PbI$_2$ and form an entirely new amorphous (lysine)$_2$PbI$_2$. To thoroughly characterize the product of such a reaction, different ratios of yellow PbI$_2$ and white lysine are mixed (Supplementary Fig. S8). The blended powders are ground for 30 minutes to ensure a complete amalgamation. Subsequently, the powders transform into a light black color after a 30-minute heating process. XRD results reveal that peaks referring to PbI$_2$/lysine can be detected at lysine to PbI$_2$ ratios of below 2 or above 2 but not at the ratio of 2:1 (Supplementary Fig. S9). For the 2:1 ratio, there are no diffraction peaks observed in the resulting material (Fig. 1d). Combining the composition analysis result (as shown in the section of preparation of (lysine)$_2$PbI$_2$), we can confirm the formation of amorphous (lysine)$_2$PbI$_2$.

X-ray photoelectron spectroscopy (XPS) is conducted to probe the changes in the elements and their chemical states in lysine, PbI$_2$, and (lysine)$_2$PbI$_2$ (Supplementary Fig. S10). All core-level peaks are assigned to C 1$s$, N 1$s$, Pb 4$f$, and I 3$d$. The peaks referred to Pb 4$f$ in

(lysine)$_2$PbI$_2$ shift towards lower energy compared to the case of PbI$_2$, which could be attributed to the interaction between lysine and Pb. Similarly, the I 3$d$ peaks exhibit a shift towards lower binding energy, while the peaks of N 1$s$ exhibit a shift towards higher energy. Meanwhile, Fourier-transform infrared spectroscopy (FTIR) spectra indicate the shift of C = O and the absence of O-H observed in (lysine)$_2$PbI$_2$ (Supplementary Fig. S11). These results indicate that the adjacent amino and carboxyl groups in lysine coordinate with PbI$_2$. The synchrotron X-ray source is further employed to probe the amorphous (lysine)$_2$PbI$_2$ and confirm structure information (Fig. 1e and Supplementary Fig. S12). R-space Extended X-ray Absorption Fine Structure (EXAFS) results confirm the formation of Pb-I, Pb-N, and Pb-O in (lysine)$_2$PbI$_2$ (Fig.1e and Supplementary Fig. S12). In R-space EXAFS data, a strong peak around 2.70 Å is assigned to the Pb-I bond, which is smaller than the distance Pb-I in PbI$_2$ (3.17 Å)[26]. The peaks centering at 1.11 Å and 1.69 Å could be attributed to the formation of Pb-N/Pb-O[27,28]. Inspired by the crystal structure of (lysine)$_2$NiCl$_2$[29], the bond information of Pb-I/Pb-O/Pb-N, and the 6-coordinate of Pb, we firmly verify the structure of amorphous (lysine)$_2$PbI$_2$ (Fig. 1f). Newly formed (lysine)$_2$PbI$_2$ demonstrates exceptional stability, maintaining its integrity for over 2000 hours without any observable changes (Supplementary Fig. S13).

The aforementioned results indicate that lysine can react with the PbI$_2$ phase, leading to the generation of stable amorphous (lysine)$_2$PbI$_2$ at the perovskite's surface/GBs. The primary feature of amorphous materials is their isotropic nature, which exhibits significantly improved compatibility with perovskite compared to crystal passivation materials like PbI$_2$ and lysine (Supplementary Figs. S14, S15). The influence of amorphous (lysine)$_2$PbI$_2$ on the performance of perovskite films is then investigated.

**Amorphous (lysine)$_2$PbI$_2$ enhanced and stabilized Perovskite**
The amorphous (lysine)$_2$PbI$_2$ exhibits a significantly better passivation effect compared to crystalline PbI$_2$ OR lysine molecule passivators. The photoluminescence (PL) mapping result in Fig. 2a and Supplementary Fig. S16 reveals that both PbI$_2$ and lysine can increase the PL intensity. However, these thin films still experience significant quenching near

the GBs. In contrast, the amo-FACsMA sample displays a consistently, highest PL intensity across almost the entire surface. Moreover, amo-FACsMA shows a narrower distribution of blue-shift PL emission peak, mainly centered at 790 nm (Supplementary Fig. S17), signifying a reduction in spontaneous radiative recombination through defect/trap states in amo-FACsMA perovskite film. Meanwhile, the time-resolved PL (TRPL) spectra in Fig. 2c show that amo-FACsMA perovskite exhibits the longest PL lifetime ($\tau$) (1.04 µs), which is far longer than $PbI_2$/lysine passivation (Supplementary Fig. S16). Enhanced PL intensity, blue-shift and narrow PL emission peak distribution, and longer PL lifetime indicate that non-radiative recombination in amo-FACsMA films is suppressed, which can be attributed to decreased trap densities. Consistently, the Urbach energy (Supplementary Fig. S18) is significantly decreased in amo-FACsMA. The smaller Urbach energy in amo-FACsMA corresponds to a lower density of trap states[30]. These results indicate that amorphous $(lysine)_2PbI_2$ can notably diminish the defects in FACsMA perovskite.

We have further investigated the defect physics of these samples using thermal admittance spectroscopy (TAS). All devices show typical temperature-dependent capacitance versus frequency ($C$–$f$) plots (Supplementary Fig. S19). The sub-gap energy is deduced from the temperature-dependent $C$–$f$ plots. Both $PbI_2$ and lysine passivators can decrease the trap energy to 0.49 eV and 0.47 eV, respectively, compared to the 0.73 eV trap energy of pristine perovskite (Supplementary Fig. S20). However, the amo-FACsMA-based device exhibits the lowest trap energy depth of 0.15 eV. Figure 2c and Supplementary Fig. S21 show the trap density deduced from the room-temperature $C$–$f$ plots, and the amo-FACsMA sample exhibits the lowest integrated trap density of $3.2 \times 10^{15}$ cm$^{-3}$ eV$^{-1}$. These results indicate a moderate passivation effect of crystal lysine and $PbI_2$ passivation agent. In contrast, the weak correlation of C-f with temperatures in amo-FACsMA-based devices suppressed influence from trap states, and hence excellent passivation.

In general, polycrystalline perovskites typically possess numerous defects including unsaturated dangling bonds (unsaturated $Pb^{2+}$ and ion vacancies)[6]. The crystalline passivators like $PbI_2$ and lysine possess the ability to passivate certain defects to a degree; however, they potentially introduce new issues, for example, lattice stress arising from alterations in ionic bond energy because of the robust interaction between crystalline passivation materials and surface/interface ions. In addition, because of the nature of crystal materials, the interface between polycrystalline perovskite and crystal passivators often hosts many defects, such as dislocations, dangling bonds, and lattice strain. These defects significantly deteriorate the PV performance. Herein, the $PbI_2$ on the surface/interface of perovskite can serve as a seed to react with lysine, thereby forming an amorphous layer with fewer dangling bonds terminated on the surface/interface of perovskite. The interface between the crystalline and amorphous materials hosts fewer dislocations, dangling bonds, and smaller strain than the interfaces present in the polycrystalline sample (crystal-crystal, crystal-air)[23,31,32]. Consequently, the amorphous $(lysine)_2PbI_2$ demonstrates remarkable passivation effects, leading to a substantial decrease in trap energy depth and defect density within the amo-FACsMA sample. These outcomes are beneficial to mitigating non-radiation recombination and enhancing PV performance.

In addition, the amorphous $(lysine)_2PbI_2$ significantly improves the stability of $PbI_2$-FACsMA perovskite films. The XRD patterns of the FACsMA and amo-FACsMA perovskite films after 2000 h continuous illumination or/and heating are comparatively presented in Fig. 2d–f. The $PbI_2$ impurity peak is distinct for the aged $PbI_2$-FACsMA perovskite film (especially under the light and thermal coupling condition), which could be attributed to the thermal decomposition of organic cations ($FA^+$ and $MA^+$) in $PbI_2$-FACsMA perovskite. Interestingly, the aged amo-FACsMA sample exhibits hardly any detectable $PbI_2$ peak. These results suggest that the amorphous $(lysine)_2PbI_2$ can suppress perovskite de-

structuring by enhancing the resistance to thermal/photo-induced decomposition of $PbI_2$-FACsMA perovskite.

In general, during the aging process of perovskites featuring $FA^+$/$MA^+$ cations, a significant quantity of $PbI_2$ becomes evident, resulting from the deprotonation of $FA^+$/$MA^+$ cations and the subsequent formation of MA/FA, accompanied by volatilization. Herein, the $NH^{3+}$ functional group in amorphous $(lysine)_2PbI_2$ can donate protons, thereby inhibiting the deprotonation of FA and MA cations. Moreover, amorphous $(lysine)_2PbI_2$ has exhibited improved compatibility and significantly reduced the lattice stress (confirmed by the GIXRD result), which favors for decrease in the escape and release of organic cation. As such, amorphous and isotropic $(lysine)_2PbI_2$ can resist the deprotonation of $FA^+$/$MA^+$ cations and also reduce the residual stress, therefore leading to improved stability of the thin film.

### Photovoltaic performances of the PSCs

Benefiting from the advantages of amorphous $(lysine)_2PbI_2$, the resulting amo-FACsMA perovskite shows much enhanced device performance. Figure 3a compares the current density-voltage (J-V) characteristics of champion PSCs based on $PbI_2$-FACsMA, and amo-FACsMA perovskites, respectively. The amo-FACsMA-based PSC exhibits an impressive PCE of 26.27% compared with 23.72% for $PbI_2$-FACsMA. We also obtained a certified PCE of 25.94% with negligible hysteresis for amo-FACsMA -based PSC (Supplementary Fig. S22). The most striking difference is the $V_{oc}$, which increases from 1.104 V in $PbI_2$-FACsMA to 1.184 in amo-FACsMA. The EQE (Fig. 3b) is similar for both devices, with a high value of over 90% in the wavelength range of 450 ~ 700 nm. Figure 3c compares the PV parameters of FACsMA and amo-FACsMA- based PSCs for 32 devices respectively, indicating that amorphous $(lysine)_2PbI_2$ also improves the device reproducibility. In addition, the amo-FACsMA-based PSCs exhibit a smaller hysteresis (Supplementary Fig. S23 and Table S1), resulting in a stabilized output power of 26.27% (Fig. 3d). We further fabricate PSCs with an effective area of 1 cm$^2$ based on these amo-FACsMA films. The champion amo-FACsMA device displays a PCE of 24.93%, which is far higher than that of $PbI_2$-FACsMA devices (~ 21.38%) (Fig. 3e and Supplementary Fig. S24 and Table S2).

The significantly improved $V_{oc}$ in the amo-FACsMA device aligns with previous photophysical measurements on the films, i.e., the amorphous $(lysine)_2PbI_2$ effectively reduces the defect density. Further measurements on the devices also reach similar conclusions. The trap-filled limiting voltage in the space-charge limited current (SCLC) measurements decreases from 0.096 V in the FACsMA device to 0.069 V in the amo-FACsMA device (Supplementary Fig. S25), indicating reduced defects upon isotropic $(lysine)_2PbI_2$ formation. This result is also consistent with previous TAS and transient photovoltage (TPV) decay results (Fig. 3f). Compared to the FACsMA device (0.14 µs), the slower $V_{oc}$ decay (0.53 µs) in amo-FACsMA device indicates a longer recombination lifetime, which can contribute to a higher $V_{oc}$[33,34].

### Shelf-life and operational stability of the PSCs

In addition to improved PV performance, the amo-FACsMA device shows significantly enhanced stability. We measure the shelf life by storing the unencapsulated devices in the dark at 25 °C in an $N_2$ glovebox. Figure 4a demonstrates a 10% decrease in the PCE of the FACsMA device after 100 days of aging, while the amo-FACsMA device exhibits minimal variation over the same aging period. We then investigate the long-term operational stability of the PSCs by aging the encapsulated devices in the ambient, using maximum power point (MPP) tracking under simulated 1-sun conditions. As shown in Fig. 4b, the amo-FACsMA-based PSCs retain over 94% of the initial PCE while the FACsMA device maintains only approximately 20% PCE after the 1000 h MPP test. Even at 85 °C, the amo-FACsMA-based PSCs also demonstrate excellent stability (Supplementary Fig. 4c).

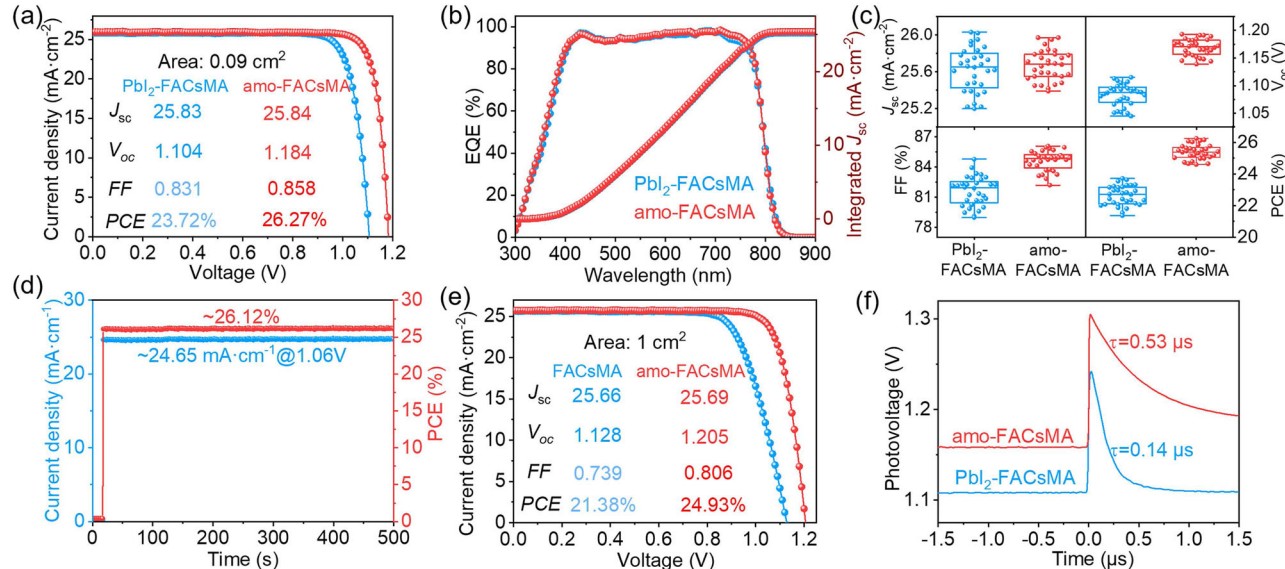

**Fig. 3 | Photovoltaic and device characterization. a** The J-V curves of the champion devices of PbI$_2$-FACsMA and amo-FACsMA with an effective cell area of 0.09 cm$^2$ in the reverse scan. **b** EQE spectra and integrated $J_{sc}$ of PbI$_2$-FACsMA and AMO-FACsMA-based PSCs. **c** The PV parameters distribution of PbI$_2$-FACsMA and amo-FACsMA PSCs from 32 devices, respectively. **d** Steady-state efficiency of amo-FACsMA PSCs. **e** J-V characteristics of PSCs based on PbI$_2$-FACsMA and amo-FACsMA PSCs with a 1 cm$^2$ effective cell area in the reverse scan. **f** TPV curves of PbI$_2$-FACsMA and amo-FACsMA-based devices.

The enhanced stability of PSCs is attributed to the suppressed perovskite de-structuring of light-absorbing layers in amo-FACsMA PSCs. In Fig. 4d, we compare the cross-section images of the perovskite layer for the PSCs based on FACsMA and amo-FACsMA after the 500 h MPP test. Contrasting the cross-section of the initial perovskite device (Supplementary Fig. S26), the perovskite layer in the FACsMA PSCs displays a softened texture with emerged new phases. Combining our previous results on the film stability, the new phase could be assigned to PbI$_2$. In contrast, the cross-sectional morphology of the amo-FACsMA device shows no discernible changes, which can be attributed to the stabilizing effect of isotropic (lysine)$_2$PbI$_2$.

We further conducted an in-depth analysis of component distribution within the perovskite layer for PSCs based on the FACsMA and amo-FACsMA perovskite films. In the FACsMA device (Fig. 4e), a notable offset component distribution is observed compared with the initial devices shown in Supplementary Fig. S27. Both MA$^+$ and FA$^+$ cations exhibit reduced intensity within the perovskite layer and are found to migrate to the Au/BCP/C$_{60}$ layer. Figure 4f illustrates the 3D MA$^+$/FA$^+$ distribution in the aged FACsMA PSCs, with a concentration of these cations at the top of the devices. These results indicate that FACsMA perovskite in the PSCs undergoes a perovskite de-structuring process due to the escape and release of MA/FA from the perovskite layer during the MPP test. In contrast, in the amo-FACsMA device, both MA$^+$ and FA$^+$ cations maintain a consistently uniform distribution before and after the stability test (Figs. 4e, f and Supplementary S27). These results are consistent with previous thin film stability results and highlight that isotropic (lysine)$_2$PbI$_2$ effectively suppresses the de-structuring of FACsMA perovskite in amo-FACsMA devices through stabilizing organic components, therefore leading to significantly enhanced operational stability.

In summary, we have developed a new amorphous (lysine)$_2$PbI$_2$ enhanced and stabilized perovskites, effectively neutralizing surface/interface defects and suppressing perovskite de-structuring. Finally, the amo-FACsMA-based PSCs achieved an impressive efficiency of 26.27%, with exceptional operational stability. This work represents a breakthrough in developing highly efficient and stable perovskite materials with amorphous passivators for diverse applications, including solar cells, light-emitting diodes, and lasers.

## Methods
### Materials
Formamidinium iodide (FAI) and methylammonium iodide (MAI) were purchased from Greatcell Solar Materials Pty Ltd. Lead (II) iodide (PbI$_2$, 99.99%) and lysine were purchased from Tokyo Chemical Industry Co., Ltd. Cesium iodide (CsI) was purchased from Thermo Fisher Scientific Co., Ltd. Methylamine hydrochloride (MACl, 99%), (4-(7H-dibenzo [c, g] carbazol-7-yl) butyl) phosphonic acid (4PADCB), C60 and BCP (> 99% sublimed) were purchased from Xi'an Polymer Light Technology Corp. N, N-dimethylformamide (DMF, anhydrous, 99.8%), dimethyl sulfoxide (DMSO, anhydrous, 99.7%), chlorobenzene (anhydrous, 99.8%) and isopropanol alcohol (IPA, 99.5%) were purchased from J&K Scientific Ltd.

### Device fabrication
Perovskite precursor solution. 1.63 M perovskite precursor solution was prepared by dissolving 21.1 mg CsI, 25.9 mg MAI, 238.3 mg FAI, 15 mg MACl, 830 mg PbI$_2$ in 1 mL mixed DMF and DMSO solvents (volume ratio: 4:1) and stirred in a N$_2$ glovebox at room temperature overnight before film fabrication.

### Perovskite solar cell fabrication
The ITO glass was washed by ultrasonication with water, acetone, and IPA sequentially and then treated with UV−Ozone for 15 min before use. 4PADCB IPA solution with a concentration of 0.3 mg/mL was spin-coated on the ITO glass at 5000 r.p.m. for 30 s, followed by annealing at 100 °C for 10 min. 50 μL perovskite precursor was then spin-coated on the 4PADCB/ITO substrate by a two-consecutive step program at 1000 rpm for 5 s and 5000 rpm for 30 s. During the second step, 150 μL CB was dripped on film 14 seconds before the end of the program. Subsequently, the samples were annealed on a hotplate at 100 °C for 30 min. For the amo-FACsMA preparation, a lysine thin film is created by spin-coating a 10 mg/mL lysine water solution onto a glass substrate, followed by annealing at 100 °C for 10 min. The resulting lysine thin films are placed onto the surface of the PbI$_2$-FACsMA films and a 50 g weight is applied to the lysine films, which are then annealed at 100 °C for 10 minutes and make the lysine diffuse into PbI$_2$-FACsMA films. Consequently, a new lysine film is replaced, and the above

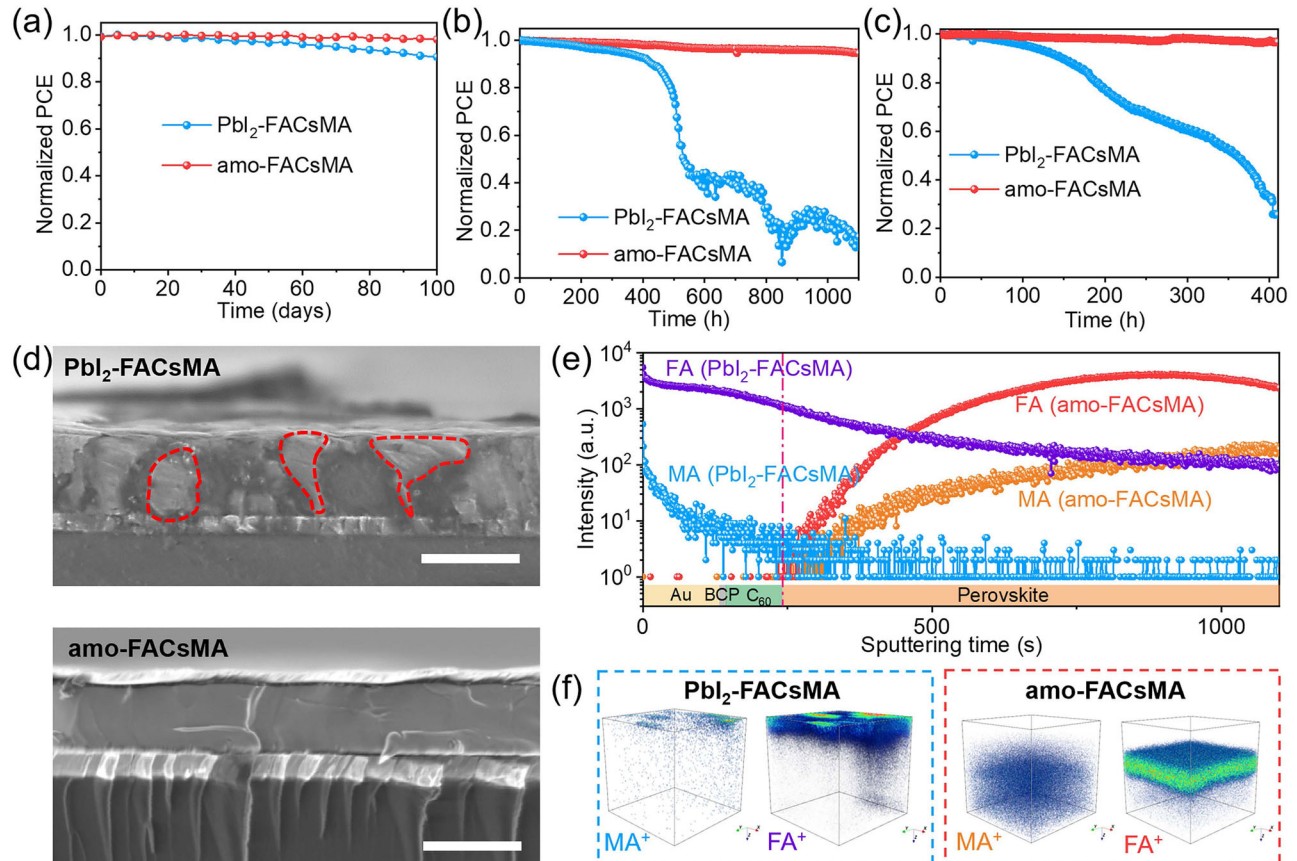

**Fig. 4 | Stability of the PbI₂-FACsMA and amo-FACsMA PSCs. a** The shelf-life stability of unencapsulated PbI₂-FACsMA and amo-FACsMA PSCs. **b** The long-term operational stability of FACsMA and amo-FACsMA PSCs. **c** The long-term operational stability of PbI₂-FACsMA and amo-FACsMA PSCs at 85 °C. **d** Cross-section images of PbI₂-FACsMA and amo-FACsMA PSCs after the 500-h operational stability test at their MPP. **e** TOF-SIMS depth profiles for the aged devices based on PbI₂-FACsMA and amo-FACsMA perovskites, Corresponding (**f**) 3D MA⁺/FA⁺ distribution in the aged PSCs.

process is repeated five times to ensure the diffusion of the lysine molecules. For comparison, in the case of pure PbI₂-FACsMA films, a blank glass is used to repeat the aforementioned process. Finally, $C_{60}$(40 nm), BCP (7 nm), and Cu (100 nm) were sequentially deposited on top of the above perovskite by thermal evaporation.

## Preparation of (lysine)₂PbI₂

The amorphous (lysine)₂PbI₂ was prepared by grinding the blended powders of 461 mg PbI₂ and 292 mg lysine for 30 min to ensure a complete amalgamation. Subsequently, a further 30-minute heating process transforms the powders into a light black color. The final amorphous product formed was dried in a vacuum. Yield = ~100%. Anal. calc. for $C_{12}H_{28}O_4N_4PbI_2$: C, 19.13; H, 3.75; N, 7.44%. Found: C, 19.17; H, 3.69; N, 7.47%.

## Film characterization

Elemental analyses for C, H, and N were performed with an Elementar Vario EL III analytical instrument. The XRD patterns were measured by an X-ray diffractometer (Empyrean, Holland PANalytical) with Cu Kα radiation ($\lambda$ = 0.154178 nm). Grazing Incidence X-ray Diffraction (GIXRD) analysis was carried out with the Rigaku SmartLab diffractometer using Cu Ka radiation. The UV-vis spectra of the films were measured using a Cary-60 UV-vis spectrophotometer. Time-resolved photoluminescence (TRPL) was measured by FLS1000 Photoluminescence Spectrometer (Edinburgh Instruments Ltd.) with a 445 nm excitation laser. The TRPL was detected at wavelength 790 nm. The PL mapping was characterized by a confocal Raman microscope (WITec) with a 532 nm excitation laser, and an emission signal was collected in the wavelength range from 765 to 815 nm with a center wavelength of 790 nm. The top-view and cross-section SEM images were performed using a Zeiss 500 field in high vacuum mode. The XPS spectra were carried out on an AXIS Ultra DLD spectrometer by using an Al-Ka X-ray source. The TOF-SIMS analysis was measured by TOF SIMS 5-100 (ION-TOF GmbH, Germany). FTIR analysis was measured by ThermoFisher iS50.

GIWAXS measurements were performed at the BL14B1 beamline of the Shanghai Synchrotron Radiation Facility (SSRF) with a beam wavelength of 0.12398 nm. Pb $L_3$-edge analysis was performed with Si (111) crystal monochromators at the BL11B beamlines at the SSRF. Before the analysis at the beamline, samples were pressed into thin sheets 1 cm in diameter and sealed using Kapton tape film. The XAFS spectra were recorded at room temperature using a 4-channel Silicon Drift Detector (SDD) Bruker 5040. Pb $L_3$-edge extended X-ray absorption fine structure (EXAFS) spectra were recorded in transmission mode. Negligible changes in the line shape and peak position of Pb $L_3$-edge XANES spectra were observed between two scans taken for a specific sample. The XAFS spectra of these standard samples (Pb foil and PbO) were recorded in transmission mode. The spectra were processed and analyzed by the software codes Athena and Artemis.

## Device characterization

J-V characteristics of the devices were measured with a Keithley 2401 source meter under the simulated AM 1.5 G illumination (100 mW·cm⁻²) using an Enlitech 3 A light source (reserve scan: 1.25 V − (− 0.1 V), forward scan: (− 0.1 V) − 1.25 V, scan rate: 0.05 V·S⁻¹). The light intensity was calibrated by an SRC2020-KG1 stand Si cell before the

test. The aperture area of non-reflective metal masks in J-V measurement are 0.09 cm² and 1.0 cm². The EQE spectra were performed by the Enlitech QE-3011 system. The TPV measurements were acquired using a DPO4104 oscilloscope, obtained through nanosecond pulse laser light incident on solar cells under short-circuit and open-circuit conditions. The dark I–V characteristics for SCLC measurements were recorded for devices with structures of $FTO/TiO_2/SnO_2/Perovskite/C_{60}/BCP/Cu$ using a Keithley 2401 source meter.

Stability test: The thermal stability of perovskite thin films was tested on a 55 °C hotplate and/or light soaking, respectively. Both the thermal and photo-stability tests were performed in an $N_2$ glove box without temperature control. We have tried to test the temperature on the surface of the cell during the continuous MPP test. The temperature of the cell is approximately ~50 °C. The encapsulated PSCs were continuously illuminated in a white LED lamp (100 mW·cm⁻²) to measure the devices' photo-stability stability.

### Trap density measurements by thermal admittance spectroscopy

A sinusoidal voltage ($V_{peak-to-peak} = 30$ mV) generated from a function generator (Tektronix AFG3000) was applied to the device. The current signal of the devices was analyzed using a lock-in amplifier (Stanford Research Systems, SR830) after being amplified through a low-noise-current preamplifier (Stanford Research Systems, SR570). The capacitance of the device was calculated based on the parallel equivalent circuit model with the amplitude and phase of the current signal obtained from the lock-in amplifier. The capacitance spectra of the device were measured using an impedance analyzer (TH2836) by scanning the frequency of the AC voltage from 0.01 to 100 kHz in logarithmic steps. The temperature of the device was controlled using a temperature-controlled probe station. The capacitance-voltage curve was obtained by measuring the capacitance as the applied DC bias voltage was scanned from −0.2 to 1.2 V, based on the capacitance spectra measured at different temperatures, the trap density distribution in energy (E.) was calculated with the following equations:

$$N_T(E\omega) = -\frac{V_{bi}}{qW}\frac{dC}{d\omega}\frac{\omega}{k_BT} \quad (1)$$

$$E_\omega = k_BTIn\left(\frac{\omega_0}{\omega}\right) = k_BTIn\left(\frac{2\pi v_0 T^2}{\omega}\right) \quad (2)$$

where $V_{bi}$ is the built-in potential and $W$ is the depletion width ($V_{bi}$, and $\omega$ are derived from capacitance-voltage measurements); $C$ is the capacitance measured at an angular frequency $w$ and temperature $T$; $k_B$ is Boltzmann's constant and $\omega_0$ is the attempt-to-escape frequency at temperature T, and $v_0$ is a temperature-independent constant obtained by fitting the characteristic frequency at different $T$ values with the equation.

### Reporting summary
Further information on research design is available in the Nature Portfolio Reporting Summary linked to this article.

## Data availability
The data supporting the findings of this study are provided in the main text and the Supplementary Information. More data are available from the corresponding author upon request.

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

## Acknowledgements

Y. W. thanks the National Science Fund for Excellent Young Scholars (Overseas), the Top Talent Project of West Lake Pearl Project, the National Natural Science Foundation of China (No. 52302315), and the talent project of ZJU-Hangzhou Global Scientific and Technological Innovation Center (No. 02170000-K02013017). D. Yang thanks the project of National Natural Science Foundation of China (No. 61721005). X. Wang thanks the project of China Huaneng Group Key R&D Program (HNKJ22-H104). T. Liu thanks the project of the Natural Science Foundation Program of Shaanxi Province (2023-JC-QN-0143). Yu Wang thanks the European Union's Horizon 2020 research and innovation program with a Marie Skłodowska-Curie grant agreement No 956270. W. Ning thanks the support from the Gusu Innovation and Entrepreneurship Leading Talent Program (ZXL2023188), the Jiangsu Key Laboratory for Carbon-Based Functional Materials and Devices (Z221311), and the Suzhou Key Laboratory of Functional Nano & Soft Materials, Collaborative Innovation Center of Suzhou Nano Science& Technology, the 111 Project.

## Author contributions

Y. Wang conceived the idea. D. Yang, Y. Wang, W. Ning supported this work. Y. Wang, Y. Wen and T. Zhang fabricated and characterized the devices. X. T. Wang and T.T. Liu carried out the PL/TRPL, PL mapping experiments. R. Zhang contributed to the GIWAX test and analysis. M. Kan contributed to the EXAFS analysis. Yu Wang actively contributed to all discussions. Y. Wang wrote the original draft and D. Yang, Y. Wang, W. Ning, T.T. Liu, L. Wan and X.T. Wang reviewed and edit the draft.

## Competing interests

The authors declare no competing interests.
