## [Peer Review File · Nature Communications]

Amorphous (lysine)₂PbI₂ Layer Enhanced Perovskite
PhotovoltaicsEditorial Note: This manuscript has been previously reviewed at another journal that is not operating a transparent peer review scheme. This document only contains reviewer comments and rebuttal letters for versions considered at *Nature Communications*.

REVIEWER COMMENTS

Reviewer #1 (Remarks to the Author):

The revision has addressed all the concerns. I am pleased with the author's response and endorse the publication of their work in its current form.

Reviewer #2 (Remarks to the Author):

In the revised version of the manuscript the authors have presented additional data from GIWAX to XRD and XPS to univocally assess the formation of the amorphous phase, previously rising. The new data corroborate their observations and clarify the diffusion mechanisms when the lysine interacts with the perovskite surface. Given the solid conclusions and the large amount of additional data which I think are appropriate to elucidate the mechanism behind, I believe the overall quality have greatly improved. The only information which would require a little bit more attention is the stability of the film and of the device. Can the author comment on that? I think the paper can after a proper second revision be potentially interesting for Nat Comm readership.

Reviewer #4 (Remarks to the Author):

In this research article, Wen et al. report a new passivation approach based on an amorphous (lysine)₂PbI₂ layer for enhancing the performance of perovskite solar cells. By reacting a PbI₂-rich perovskite layer with a separately prepared solid lysine film, the authors synthesize an amorphous (lysine)₂PbI₂ layer at surface/grain boundaries in the perovskite film. This amorphous passivation layer suppresses the interface defect density, reduces lattice stress in the perovskite, and enhances the stability of the perovskite film. As a result, an impressive power conversion efficiency of more than 26% has been achieved.

In this revision, the authors added new evidence to strengthen the experimental explanations of the solid-diffusion reaction mechanism between PbI₂ and lysine as well as the (lysine)₂PbI₂ working principle in enhancing the performance of perovskite solar cells. The GIXRD data supports the elimination of residual stress within the perovskite lattice by introducing the amorphous passivation layer. The PL intensity and lifetime measurements justify the superior passivation effect of (lysine)₂PbI₂ than stay-alone PbI₂ and lysine passivation. Thermal admittance spectroscopy (TAS) measurements further quantify the impact of suppressing defects in the perovskite films.

Overall, this revised manuscript has adequately addressed the comments from the last round of review and is in good shape for publication. Some minor comments are provided below to further polish the manuscript for high quality publication.

[1] A color scale and elements used for the ToF-SIMS measurements should be included in Supplementary Fig. S6 to help readers understand the figure.

[2] Supplementary Fig. S10 shows both Pb 4f and I 3d shift toward lower binding energies after reacting PbI₂ with lysine, contradicting the main text, which claims Pb 4f shifts toward higher energy. Please double-check the results.

[3] The cross-sectional SEM images of PbI₂-FACsMA films shown in Fig. 1b and Supplementary Fig. S27 are quite different. It seems that the authors attempt to exaggerate the issue in PbI₂-FACsMA film (titled grains, high density of PbI₂) using Fig. 1b. Please check the data and use regular/representative data rather than extremes.

[4] The cell temperature of the MPPT measurement reported in Fig. 4b needs to be specified. The 85 °C stability measurement data (Supplementary Fig. S26) can be merged into Fig. 4 of the main manuscript to convey better the message of the impact of the amorphous (lysine)₂PbI₂ on suppressing the thermal decomposition of the perovskite film.

Replies to the comments

Reviewer #2 (Remarks to the Author):

In the revised version of the manuscript the authors have presented additional data from GIWAX to XRD and XPS to univocally assess the formation of the amorphous phase, previously rising. The new data corroborate their observations and clarify the diffusion mechanisms when the lysine interacts with the perovskite surface. Given the solid conclusions and the large amount of additional data which I think are appropriate to elucidate the mechanism behind, I believe the overall quality have greatly improved. The only information which would require a little bit more attention is the stability of the film and of the device. Can the author comment on that? I think the paper can after a proper second revision be potentially interesting for Nat Comm readership.

Re: Thank you for acknowledging our previous revision. We agree with you that we should pay more attention to the stability of the film materials and of the device. The amorphous (lysine)₂PbI₂ significantly improves the stability of PbI₂-FACsMA perovskite films. Considering the real operation of the perovskite solar cells, we investigated the stability of perovskite films under thermal or light conditions. The XRD patterns of the FACsMA and amo-FACsMA perovskite films after 2000 h continuous illumination or/and heating are comparatively presented in Fig 2d-2f. The PbI₂ impurity peak is distinct for the aged PbI₂-FACsMA perovskite film (especially under the light and thermal coupling condition), which could be attributed to the decomposition of organic cations (FA⁺ and MA⁺) in PbI₂-FACsMA perovskite. Interestingly, the aged amo-FACsMA sample exhibits hardly any detectable PbI₂ peak, which can be contributed that the NH₃⁺ functional group in amorphous (lysine)₂PbI₂ can donate protons, thereby inhibiting the deprotonation of FA and MA cations.

Fig.1 XRD results of perovskite films after (d) 55°C heating, (e) 1 sun illumination, and (f) 55°C heating + 1 sun illumination stability test in a glove box, the diamond represents PbI₂.

C, 85% RH) and analysis the aged devices via TOF-SIMS. The in-depth analysis of component distribution by TOF-Sims further confirm that, in the case of amo-FACsMA devices, amorphous (lysine)₂PbI₂ significantly suppress the decomposition of organic components within the perovskite layer (Fig. R1). In the FACsMA device, a notable offset component distribution is observed compared with the initial devices. Both MA⁺ and FA⁺ cations exhibit reduced intensity within the perovskite layer and are found to migrate to the Au/BCP/C₆₀ layer. Fig. 4d illustrates the 3D MA⁺/FA⁺ distribution in the aged FACsMA PSCs, with a concentration of these cations at the top of the devices. These results indicate that the escape and release of MA/FA from the perovskite layer occurs during MPP test. In contrast, in the amo-FACsMA device, both MA⁺ and FA⁺ cations maintain a consistently uniform distribution before and after the stability test (Fig. R1). Such result well consistent with our previous thin film test result (Fig. 1e-1f). All of these results highlight that amorphous (lysine)₂PbI₂ effectively suppress the de-structuring of FACsMA perovskite through stabilizing organic components, therefore leading to significantly enhanced materials and device's operational stability.

Fig. R1 (a) TOF-SIMS depth profiles for the devices based on PbI₂-FACsMA and amo-FACsMA perovskites before MPP tests. Corresponding (b) 3D MA⁺/FA⁺ distribution in the PSCs based on PbI₂-FACsMA and amo-FACsMA perovskites before MPP tests. (c) TOF-SIMS depth profiles for the aged devices based on PbI₂-FACsMA and amo-FACsMA perovskites, Corresponding (d) 3D MA⁺/FA⁺ distribution in the aged PSCs.

In general, during the aging process of perovskites featuring FA⁺/MA⁺ cations, a significant quantity of PbI₂ becomes evident, resulting from the deprotonation of FA⁺/MA⁺ cations and the subsequent formation of MA/FA, accompanied by volatilization (i.e., the called organic decomposition). Through our previous comprehensive investigation, we can confirm that lysine can interact PbI₂ to form amorphous (lysine)₂PbI₂. The NH₃⁺ functional group in amorphous (lysine)₂PbI₂ can donate protons, thereby inhibiting the deprotonation of FA and MA cations. Moreover, as confirmed by GIXRD result

in our previous revision version, amorphous $(\text{lysine})_2\text{PbI}_2$ have exhibited improved compatibility and significantly reduce the lattice stress, which is different from the case of those crystal passivators. The residual stress in thin film always favors for increase the escape and release of organic cation, as well as the decomposition. In our work, amorphous and isotropic $(\text{lysine})_2\text{PbI}_2$ not only resists the deprotonation of FA^+/MA^+ cations, but also reduce the residual stress. As such, the decomposition of organic components and de-structuring of perovskite are successfully suppressed, leading to improved stability of the thin film and device.

Reviewer #4 (Remarks to the Author):

In this research article, Wen et al. report a new passivation approach based on an amorphous $(\text{lysine})_2\text{PbI}_2$ layer for enhancing the performance of perovskite solar cells. By reacting a PbI_2 -rich perovskite layer with a separately prepared solid lysine film, the authors synthesize an amorphous $(\text{lysine})_2\text{PbI}_2$ layer at surface/grain boundaries in the perovskite film. This amorphous passivation layer suppresses the interface defect density, reduces lattice stress in the perovskite, and enhances the stability of the perovskite film. As a result, an impressive power conversion efficiency of more than 26% has been achieved.

In this revision, the authors added new evidence to strengthen the experimental explanations of the solid-diffusion reaction mechanism between PbI_2 and lysine as well as the $(\text{lysine})_2\text{PbI}_2$ working principle in enhancing the performance of perovskite solar cells. The GIXRD data supports the elimination of residual stress within the perovskite lattice by introducing the amorphous passivation layer. The PL intensity and lifetime measurements justify the superior passivation effect of $(\text{lysine})_2\text{PbI}_2$ than stay-alone PbI_2 and lysine passivation. Thermal admittance spectroscopy (TAS) measurements further quantify the impact of suppressing defects in the perovskite films.

Overall, this revised manuscript has adequately addressed the comments from the last round of review and is in good shape for publication. Some minor comments are provided below to further polish the manuscript for high quality publication.

[1] A color scale and elements used for the ToF-SIMS measurements should be included in Supplementary Fig. S6 to help readers understand the figure.

Re: Thanks for your comments. We have included the color scale and elements in the revised supplementary Fig. S6.

Revised Supplementary Fig. S6 Lysine components analysis. Two-dimensional image (100 μm × 100 μm) of total depth accumulation (up), and cross-section image (down) in (a) PbI₂-FACsMA and (b) amo-FACsMA perovskite films.

[2] Supplementary Fig. S10 shows both Pb 4f and I 3d shift toward lower binding energies after reacting PbI₂ with lysine, contradicting the main text, which claims Pb 4f shifts toward higher energy. Please double-check the results.

Re: We deeply apologize for our previous input error. Now, we have revised the main text as “The peaks referred to Pb 4f in (lysine)₂PbI₂ shift towards lower energy compared to the case of PbI₂, which could be attributed to the interaction between lysine and Pb.”

[3] The cross-sectional SEM images of PbI₂-FACsMA films shown in Fig. 1b and Supplementary Fig. S27 are quite different. It seems that the authors attempt to exaggerate the issue in PbI₂-FACsMA film (titled grains, high density of PbI₂) using Fig. 1b. Please check the data and use regular/representative data rather than extremes.

Re: Thanks for your comments. Due to the effects of film/device cross-section preparation, deviations always occur in the sample of film and device. Now, we have re-prepared and re-tested the cross-section of the device, as shown in Revised Supplementary Fig. S26. The cross-sections of thin film and device show similar PbI₂ density.

Fig. 2b-2c Top-surface and cross-section images of (b) PbI_2 -FACsMA and (c) amo-FACsMA perovskite films, the red dash box represents PbI_2 .

Supplementary Fig. S26 Initial cross-section images of (a) PbI_2 -FACsMA and (b) amo-FACsMA PSCs before operational stability test, scale bars represent 1 μm .

[4] The cell temperature of the MPP measurement reported in Fig. 4b needs to be specified. The 85 C stability measurement data (Supplementary Fig. S26) can be merged into Fig.4 of the main manuscript to convey better the message of the impact of the amorphous $(\text{lysine})_2\text{PbI}_2$ on suppressing the thermal decomposition of the perovskite film.

Re: Thanks for your constructive comments. Our experiments were conducted at room temperature without temperature control. Now, we have tried to test the temperature on the surface of the cell during the continuous MPP test. The temperature of the cell is approximately $\sim 50^\circ\text{C}$. Additionally, we have included this information into the experimental section. Additionally, we have merged previous Supplementary Fig. S26 into Fig. 4 (i.e., New Fig 4c).

Revised Fig. 4 | Stability of the $\text{PbI}_2\text{-FACsMA}$ and amo-FACsMA PSCs. (a) The shelf-life stability of unencapsulated $\text{PbI}_2\text{-FACsMA}$ and amo-FACsMA PSCs. (b) The long-term operational stability of FACsMA and amo-FACsMA PSCs. (c) The long-term operational stability of $\text{PbI}_2\text{-FACsMA}$ and amo-FACsMA PSCs at 85 °C. (d) Cross-section images of $\text{PbI}_2\text{-FACsMA}$ and amo-FACsMA PSCs after the 500-h operational stability test at their MPP. (e) TOF-SIMS depth profiles for the aged devices based on $\text{PbI}_2\text{-FACsMA}$ and amo-FACsMA perovskites, Corresponding (f) 3D MA^+/FA^+ distribution in the aged PSCs.

REVIEWERS' COMMENTS

Reviewer #2 (Remarks to the Author):

The authors have reported additional data which further prove and confirm the main findings. For this reason I would consider the work for publication.

Reviewer #4 (Remarks to the Author):

The authors adequately addressed the reviewers' questions, and the manuscript is now in good shape for publication.